An updated assessment of Symbiodinium spp. that associate with common scleractinian corals from Moorea (French Polynesia) reveals high diversity among background symbionts and a novel finding of clade B

Rouzé Héloïse heloise.rouze@gmail.com 1
Lecellier Gaël J. 1 2 6
Saulnier Denis 3
Planes Serge 1
Gueguen Yannick 4
Wirshing Herman H. 5
Berteaux-Lecellier Véronique 1 6
1 PSL CRIOBE USR3278 CNRS-EPHE-UPVD, Labex CORAIL , Papetoai , Moorea , French Polynesia
2 Université de Paris Saclay, Departement de Biologie, Versailles-Saint Quentin , Paris , Versailles Cedex , France
3 UMR241 EIO Ifremer-ILM-IRD-UPF, Labex CORAIL , Taravao , French Polynesia
4 UMR5244 IHPE, CNRS-Ifremer-UM-UPVD, Université de Montpellier , Montpellier , France
5 Department of Invertebrate Zoology, Smithsonian National Museum of Natural History , Washington, D.C. , USA
6 Current affiliation:  UMR250/9220 ENTROPIE IRD-CNRS-UR, Labex CORAIL, Promenade Roger-Laroque , Noumea cedex, New Caledonia , France
Pochon Xavier
Electronic publication date: 2017 Jan 5
Publication date: 2017
Volume: 5
Electronic Location ID: e2856
Received 2016 Apr 20; Accepted 2016 Dec 2
Copyright: ©2017 Rouzé et al.
Copyright year: 2017
Copyright holder: Rouzé et al.
License: This is an open access article distributed under the terms of the Creative Commons Attribution License, which permits unrestricted use, distribution, reproduction and adaptation in any medium and for any purpose provided that it is properly attributed. For attribution, the original author(s), title, publication source (PeerJ) and either DOI or URL of the article must be cited.
License URL: https://creativecommons.org/licenses/by/4.0/

Keywords: Corals, French polynesia, Clade B, Symbiodinium, qPCR, Flexibility, Generalist, Faithful clade

Funding: Proscience and TeMana o teMoana (French Polynesia) associations Délégation à la Recherche of French Polynesia Ministère de l’Outre-Mer HR was supported by Proscience and TeMana o teMoana (French Polynesia) associations. Additional funding was provided by the Délégation à la Recherche of French Polynesia, the Ministère de l’Outre-Mer and the contrat de projet Etat-Polynésie française. The funders had no role in study design, data collection and analysis, decision to publish, or preparation of the manuscript.

==============================
The adaptative bleaching hypothesis (ABH) states that, depending on the symbiotic flexibility of coral hosts (i.e., the ability of corals to “switch” or “shuffle” their algal symbionts), coral bleaching can lead to a change in the composition of their associated Symbiodinium community and, thus, contribute to the coral’s overall survival. In order to determine the flexibility of corals, molecular tools are required to provide accurate species delineations and to detect low levels of coral-associated Symbiodinium. Here, we used highly sensitive quantitative (real-time) PCR (qPCR) technology to analyse five common coral species from Moorea (French Polynesia), previously screened using only traditional molecular methods, to assess the presence of low-abundance (background) Symbiodinium spp. Similar to other studies, each coral species exhibited a strong specificity to a particular clade, irrespective of the environment. In addition, however, each of the five species harboured at least one additional Symbiodinium clade, among clades A–D, at background levels. Unexpectedly, and for the first time in French Polynesia, clade B was detected as a coral symbiont. These results increase the number of known coral-Symbiodinium associations from corals found in French Polynesia, and likely indicate an underestimation of the ability of the corals in this region to associate with and/or “shuffle” different Symbiodinium clades. Altogether our data suggest that corals from French Polynesia may favor a trade-off between optimizing symbioses with a specific Symbiodinium clade(s), maintaining associations with particular background clades that may play a role in the ability of corals to respond to environmental change.

Introduction

The foundation of coral reefs is based on the symbiotic association between scleractinian corals and dinoflagellates in the genus Symbiodinium. Molecular phylogenetic analyses currently subdivide Symbiodinium into nine clades (A–I), each divided further into sub-clades or types based on various molecular techniques (reviewed in Pochon, Putnam & Gates, 2014). However, corals most commonly associate with Symbiodinium in clades A–D (reviewed in Baker, 2003), and, in rare cases, with members of clades F and G (LaJeunesse, 2001; Putnam et al., 2012; Lee et al., 2016). Symbiodinium is assumed to provide up to 95% of the energy required for coral metabolic activities (Muscatine & Porter, 1977; Davy, Allemand & Weis, 2012), mostly due to their photosynthetic activity (i.e., production of carbohydrates). In return, the algae benefit by receiving a protected habitat from predation, and a source of inorganic nutrients derived from the host’s metabolism. However, this symbiosis can break down, depending on the degree of stress tolerance of either partner, in response to various stressors that may include natural and/or anthropogenic sources (e.g., increasing seasurface temperatures, ocean acidification, and sedimentation; Pandolfi et al., 2011).

The overall fitness of a coral colony depends on the biological and functional traits of the various organisms that comprise the coral holobiont, (i.e., the coral host, its Symbiodinium assemblages (Mieog et al., 2009b), and other associated microorganisms (e.g., bacteria) (Neave et al., 2016)). Moreover, some coral holobionts display different sensitivities to environmental conditions, which can correlate with specific biological characteristics such as morphology (Van Woesik et al., 2011). For example, the massive coral Porites predominately associates with a thermally tolerant Symbiodinium, type C15 (Fitt et al., 2009; Fabricius et al., 2011), and has been shown to exhibit increased resistance to environmental stressors such as temperature anomalies and experience lower mortality and/or bleaching rates compared to those observed for branching corals such as Acropora and Pocillopora (Penin, Vidal-Dupiol & Adjeroud, 2012). To date, both in situ (e.g., Rowan et al., 1997; Baker, 2003; Berkelmans & Van Oppen, 2006; Sampayo et al., 2008) and in vitro physiological studies (e.g., Banaszak, 2000; Kinzie et al., 2001; Hennige et al., 2009) suggest that Symbiodinium species are characterized by intrinsic physiological properties that enable them to be differentially suited for various environmental conditions.

Spatial partitioning of different Symbiodinium clades may occur at micro-scales within a single coral colony depending on solar irradiance, or among individual colonies across different depths (Rowan et al., 1997; Kemp et al., 2015). In addition, coral-Symbiodinium associations may be diverse, and can include either mono or multi-clade associations (Fabina et al., 2012; Silverstein, Correa & Baker, 2012). Moreover, these assorted Symbiodinium assemblages have been described in different coral colonies from the same species (Cunning, Glynn & Baker, 2013), during coral ontogeny (Abrego, Van Oppen & Willis, 2009; Little, Van Oppen & Willis, 2004), and/or in ‘normal’ vs. ‘stressful’ environmental conditions (e.g., seawater temperature anomalies; Berkelmans & Van Oppen, 2006). Symbiodinium in clade D have been identified as the predominant algal symbiont in resistant coral colonies during and after massive bleaching events, and/or, more generally, in reefs exposed to local stressors such as sedimentation and eutrophication (Van Oppen et al., 2001; Ulstrup & Van Oppen, 2003; LaJeunesse et al., 2010; LaJeunesse et al., 2014; Cooper et al., 2011). These observations highlight the importance of coral-Symbiodinium associations with respect to thermo-tolerance (Berkelmans & Van Oppen, 2006; Stat, Carter & Hoegh-Guldberg, 2006; LaJeunesse et al., 2009). Consequently, it has been proposed that corals with flexible associations with various Symbiodinium clades (or types), such as those resulting in a range of host-Symbiodinium associations, may contain an ecological advantage in the context of environmental change, a concept that is encapsulated in the ‘Adaptive Bleaching Hypothesis’ (ABH) (Buddemeier & Fautin, 1993).

The ABH asserts that there is potential for rapid ‘adaptation’ of corals facing stressful conditions by a dynamic modification of their Symbiodinium community composition either by (i) the acquisition of resistant Symbiodinium clades from free algae present in the environment (i.e., ‘switching’) or (ii) repopulation by background pre-existing resistant Symbiodinium clades (i.e., ‘shuffling’). Therefore, in the context of the ABH, coral flexibility (the ability of a coral species to associate with multiple Symbiodinium types) is of the utmost importance, and has led to the characterization of coral hosts as either ‘specialists’ (associating with a specific Symbiodinium clade) or ‘generalists’ (associating with multiple Symbiodinium clades) (Fabina et al., 2012; Putnam et al., 2012; Silverstein, Correa & Baker, 2012).

The development of molecular tools with highly sensitive detection capacities such as real-time quantitative PCR (qPCR), which is up to 1,000 times more sensitive than conventional methods (e.g., cloning, DGGEs, RFLP) (Mieog et al., 2007), allows for the detection of background symbionts (in addition to the dominant symbionts), and provides a measurable degree of host flexibility among corals (Silverstein, Correa & Baker, 2012). As a result, some studies have suggested that corals may be more flexible than previously thought (Mieog et al., 2007; Mieog et al., 2009a; Silverstein, Correa & Baker, 2012). The goal of this study was to investigate the degree of flexibility in host-symbiont partnerships among particular coral species from the under-explored Moorea island in French Polynesia using qPCR. Coral flexibility, considered here as the ability of a coral species to associate with multiple Symbiodinium clades or types in different proportions, and which represents one of the main conditions of the ABH, was tested. The presence of Symbiodinium clades A–F was quantified among five coral species, Acropora cytherea, Acropora pulchra, Pocillopora damicornis, Porites rus and Pavona cactus. Although each coral species displayed a high degree of specificity to a dominant Symbiodinium clade, all of the coral species hosted multiple Symbiodinium clades in low abundance, including partnerships never recorded in French Polynesia.

Materials & Methods

Choice of coral species

Five coral species, chosen among the most common scleractinian coral genera from the Pacific: Pocillopora (P. damicornis type β sensu Schmidt-Roach et al., 2014 Genbank references KY110998 –KY111024), Acropora (A. cytherea and A. pulchra), Porites (P. rus) and Pavona (P. cactus), were collected from a fringing reef with a depth 0.5–2.0 m off Moorea island in French Polynesia (17°30′9S, 149°50′9W) (Fig. 1). These five coral species display different biological traits, and were among corals characterized as having varying resistance during severe local bleaching events in 2002 and 2007 (Penin, Vidal-Dupiol & Adjeroud, 2012). Acropora is considered as the “sentinel” coral genus, described as having high sensitivity to environmental stressors (e.g., McClanahan et al., 2007; Penin et al., 2007; Penin, Vidal-Dupiol & Adjeroud, 2012). Conversely, the genus Porites was chosen for its high resistance to stress (e.g., Kayal et al., 2012; Penin, Vidal-Dupiol & Adjeroud, 2012), living in a wide range of habitats around the island (e.g., sedimentary bays). Finally, the last two genera, Pocillopora and Pavona, were chosen because they are considered having intermediate degrees of sensitivity (Penin, Vidal-Dupiol & Adjeroud, 2012).

Figure 1 Map of Moorea island (Archipelago of society, French Polynesia) and the locations of the fringing reefs studied (black circles).

Locations (FG) with an “x” indicate the sites investigated previously by Putnam et al. (2012). Vaiare (Va), Teavaro (Te), Maharepa (Ma), Tiahura (Ti) and Linareva (Li).

All of the coral species were sampled during the dry season between August and October 2012, P. damicornis (N = 27), P. rus (N = 21) and A. cytherea (N = 16) were sampled in greater proportions compared to A. pulchra (N = 6) and P. cactus (N = 7). Sampling was performed among five contrasting fringing reefs from the lagoon of Moorea island: Mahareapa (Ma) and Vaiare (Va) are exposed to anthropogenic influence, and Teavaro (Te), Linareva (Li) and Tiahura (Ti) are more isolated from human activities (Nahon et al., 2013; Rouzé et al., 2015).

DNA extraction

Small coral fragments (0.5–1 cm3) were sampled at several areas from the top of each coral colony, placed directly into a tube underwater, and immediately transferred at the surface into a new 1.5 mL centrifuge tube containing 80% ethanol. All samples were stored at −20 °C until DNA extraction. Prior to extraction, all of the ethanol was discarded and the coral sample gently rinsed with sterile freshwater to eliminate all traces of mucus. This allows for better targeting of Symbiodinium present in the host tissues.

Total coral DNA (i.e., Symbiodinium, polyps, and associated micro-organisms) was extracted using a CTAB-based extraction protocol adapted from Mieog et al. (2009a). To increase the efficiency of DNA extraction, coral samples were incubated in 600 µL of extraction buffer CTAB 2% (2% CTAB, 1.4 M NaCl, 20 mM EDTA pH 8, 100 mM Tris–HCl pH 8 and 20 µg/mL proteinase K). They were then exposed to 3 cryo-shock cycles (5 min in liquid nitrogen following by 10 min at ambient temperature), and incubated at 60 °C overnight while rotating. Next, the CTAB buffer was recovered and placed into a new tube containing 600 µL of chloroform/iso-amyl alcohol (24:1 vol/vol). The resulting solution was mixed thoroughly and centrifuged for 15 min at 12,000 g (4 °C). The aqueous phase was then transferred into a new tube and mixed with 600 µL of isopropanol at 0 °C, and incubated for 20 min at −20 °C. After a new round of centrifugation, the supernatant was discarded and the pellet rinsed with 500 µL of 70% ethanol. After a final centrifugation of 10 min at 12,000 g, the ethanol was removed and the DNA pellet air-dried before dilution in 100 µL sterile water (Sigma). All DNA samples were then stored at −40 °C.

qPCR assays

Primer set assessment

Six primer sets optimized for the amplification of nuclear ribosomal 28S in Symbiodinium clades A–F (Yamashita et al., 2011), and one coral-specific 18S  primer set for the coral host (i.e., polyps) were used. The 18S coral host primers (univPolyp-18SF: 5′-ATCGATGAAGAACGCCAGCCA-3′ and univPolyp-18SR: 5′CAAGAGCGCCATTTGCGTTC-3′) were designed with Primer 3 (Untergasser et al., 2012) from the 18S rDNA sequence alignment (276 sequences) of 18 coral species that are among the most abundant genera found in French Polynesia (Porites spp., Pocillopora spp., Acropora spp., Montipora spp., and Povona spp.) as well as Symbiodinium clades as negative controls.

The quality of the different primer sets for qPCR was confirmed using the evaluation of indicators of specificity and efficiency. Firstly, the specificity of the symbiont and host primer sets were verified with DNA from cultured Symbiodinium strains (available from the BURR Collection (http://www.nsm.buffalo.edu/Bio/burr/); clade A: CasskB8 and Flap1, B: Pe and Flap2, C: Mp, D: A001 and A014, E: RT383 and F: Sin and Pdiv44b), and with coral DNA from various species (Acropora: A. pulchra, A. cytherea, A. hyacynthus; Pocillopora: P. damicornis, P. verrucosa, P. meandrina; Porites rus; Pavona cactus; Montipora spp. and Fungia spp.). A percentage of specificity (Sp: expected with target/unexpected with non-target region) of the symbiont-specific primer sets was calculated according to the formula: Sp =1 − ∑ (100/2(Cti−Ctx)), where Cti and Ctx are Ct obtained from a specific primer set (Cti) and from other primer sets (Ctx) on the same target DNA sample. Secondly, the efficiency of the different primer sets was estimated from the standard curve method based on the log-linear regression of the Ct values with 10-fold serial dilutions of the DNA over 7 concentrations. For both Symbiodinium and the coral host, the matrix for dilution was based on a series of known DNA concentrations. In order to mimic multiclade associations and/or the DNA complexity, the matrix was performed by a mixture of several DNA extracts equally concentrated for Symbiodinium (70 ng of each clades A–F; one reference strain per clade; clade A: CasskB8, B: Pe, C: Mp, D: A001, E: RT383 and F: Sin), and the coral host (on 1/mixture multi-specific: 50 ng of the ten coral species mentioned above or 2/mixture mono-specific: 50 ng of five DNAs from the same coral species for P. damicornis, P. rus or A. cytherea). Additionally, for Symbiodinium the standard curve method was applied on a series of known 28S rDNA copy numbers (amplified DNA for clades A–F; Supplemental Information 1), or a series of known cell densities of clade A, C and D isolated from the coral hosts (clade B was not available; Supplemental Information 1). Percentage of efficiency (100% of efficiency indicates that the amount of PCR product doubles during each cycle) was the ratio of the observed slope and the expected slope (−3.322) of the log-linear regression. In addition, the standard curves of efficiency for each corresponding primer set denoted sensitivity, which corresponded to the threshold of Ct ranges to ensure an accurate amplification (i.e., the limits of the detectable log-linear range of the PCR).

Quantification of Symbiodinium in coral hosts

In order to compare Symbiodinium clades amount between different coral samples, the raw number of 28S copies of each Symbiodinium clade (from A to F) was normalized within coral hosts to evaluate the Symbiodinium densities per sample. For each coral DNA sample, a value of polyp unit was estimated by the 18S copy quantification using the standard curve equation Fig. S1b in order to normalize the quantification of the Symbiodinium clades in 28S copy number, or in cell number per unit of 18S polyp.

All qPCR assays were conducted on a MX3000 Thermocycler (Stratagene) using SYBR-Green. Each reaction was performed, in a final volume of 25 µL containing: 12.5 µL of Brillant® SYBR Green Master Mix reagent, 2.5 µL of both reverse and forward primers diluted at the concentration of 4 µM, and 10 µL of DNA at various concentrations for standard curve analysis or at 1 ng.µL−1 for field sample analysis. The following run protocol was performed: 1 cycle of pre-incubation of 10 min at 95 °C; 40 cycles of amplification: 30 s at 95 °C, 1 min at 60 °C or 64 °C for Symbiodinium and coral host respectively, and 1 min at 72 °C; and a final step, for melting temperature curve analysis, of 1 min at 95 °C, 30 s at 60 °C and 30 s at 95 °C. Each sample was analysed twice on the same plate, as one technical replicate, and averaged when the variation between both Ct values was not exceeding 1 (if not, samples were re-processed until ΔCt ≤ 1). An interplate calibrator (i.e., positive control with known concentrations and Ct values: mixture of DNA from Symbiodinium clades A–F), tested in triplicate (one technical replicate), was added to each plate to calibrate Ct values (performed manually on the MxPro software to set the fluorescent threshold to a fixed Ct value) among different plates of coral DNA samples. Positive amplifications were taken into account only when both technical replicates produced Ct values inferior to the estimated threshold ranges (i.e., limit sensitivity to ensure an accurate quantification; Table S1) after correction with the interplate calibrator. In addition, all melting curve analyses ensured the specificity of the amplifications (Table S1). For new partnerships between Symbiodinium clade(s) and coral species, we further purified the qPCR products (∼100 bp) using QiaEx II Gel Extraction Kit (Qiagen GmbH, Hilden, Germany) and sequenced in both directions (GATC Biotech, Köln, Cologne, Germany).

Statistical analysis

All statistical analyses were performed using R version 3.3.1 software (R Core Team, 2013). For each Symbiodinium clade, positively quantified in coral DNA, the symbiont/host ratio (i.e., S/H ratio) were log + 1 transformed for further analyses. Slopes, intercepts, and the Pearson correlation coefficient (R2) were evaluated by pairwise comparisons with Student’s t-tests using R package RVAideMemoire version 0.9–27 (Hervé, 2013).

Discriminant analysis of principal components (DAPC) on S/H ratios, available for the 5 coral species, was performed in R package ade4 version 1.7–4 (Dray & Dufour, 2007) in order to characterize their preferential endosymbiotic assemblages and densities. Therefore, the discrimination represented by ellipses was applied through the coral species as factor.

Results

Validation and optimization of qPCR assays

For all clade-specific primer sets, the specificity of each qPCR assay was greater than 98% (Tables S1 and S2), and was characterized by a unique melting temperature (Table S1), confirming the high accuracy of each primer set to its targeted sequence. All clade-specific primers yielded a good fit linear regression with similar efficiencies close to the desired aim of 100% (95–101%; Table S1), strong linear correlations (R2 > 0.985; Fig. S1) between Ct and concentrations of DNA template, and no significant differences among slopes. This indicates that the increase in clade-specific Symbiodinium quantity is directly proportional to the number of amplification cycles regardless of whether the tests were performed on DNA from either Symbiodinium culture strains (Table S1 and see Yamashita et al., 2011), purified PCR products (Fig. S1a), or from counted Symbiodinium cells (Fig. S2). The sensitivity of the clade-specific primers allowed two groups of primer sets to be distinguished. Pairwise comparisons of the intercepts (Student’s t-test, p < 0.05) between the standard regression lines of 28S amplicons (Fig. S1a) showed earlier detection of the primers specific to clades A, B, E and F (i = 16.36 ± 0.39; Fig. S1a) when compared with the clade-specific primers to clades C and D (i = 19.83 ± 0.27; Fig. S1a). From the Symbiodinium cell extraction, clade D sensitivity was significantly different from clades A and C (Student’s t-tests, pairwise comparisons of slopes: D/A P < 0.005 and D/C P = 0.104; intercepts: D/A P < 0.001 and D/C P = 0.015; Fig. S2). The threshold of 28S copy number estimation for each clade A–F, evaluated by the absolute quantification of Symbiodinium clades, was effective under 200 copies of the gene (Table S1 and Fig. S1a).

Similarly, the specificity of the coral-specific primer set was confirmed with positive amplifications from 10 coral species and no amplifications with Symbiodinium DNA. In addition, the amplification of multi (mixture of 10 coral species) vs. mono-specific (mixture of P. rus, P. damicornis or A. cytherea) mixes with the coral-specific primer set yielded a good fit linear regression with similar efficiencies that were close to the desired aim of 100% (101%; Table S1), contained strong linear correlations (R2 > 0.99; Fig.  S1b) between Ct and concentration of DNA template, and demonstrated no significant differences among linear correlation slopes and intercepts (Student’s t-tests, pairwise comparisons among the 4 DNA mixes: P > 0.05). In order to consider the higher complexity of multi-partner coral DNA, we performed and used analyses on multi-specific mixes of Symbiodinium and coral hosts to quantify the different Symbiodinium clades in coral DNA samples.

Diversity and flexibility of dominant vs. background Symbiodinium clades

Symbiodinium clades A, C and D (among the tested clades A–F) were detected at least once in association with each of the five coral species studied, except for P. cactus which was never found associated with clade A (Fig. 2). The quantification of these clades either by 28S copy number or by cell density displayed similar orders of magnitudes when present, whatever the species (Figs. 2A and 2B). For some coral species, this represents novel associations for corals from Moorea: clade C for both Acropora species, A. cytherea and A. pulchra, clade D for P. cactus, and clades A and D for P. rus (Table 1). The corresponding 28S sequences for these novel coral-Symbiodinium partnerships revealed the presence of lineages within sub-clades: A13, C15, C1, and D1 (Table 1; Fig. S3 and Data S2). In addition, Symbiodinium clade B was detected in P. damicornis (N = 2; Fig. 2), albeit in low abundances equivalent to 26 and 183 copies of 28S (4.25 and 6.21 in log + 1, respectively Fig. 2A). However, no relationship was available to estimate this clade’s cell number. The presence of clade B was confirmed by a match to a sequence within the sub-clade B1 (PDAM2_Moo; Fig. S3 and Data S2). Two slightly different profiles in temperature melting curves were obtained with clade C amplification for P. rus. Their sequences showed that each profile corresponded to two distinct lineages within sub-clades (Fig. S3 and Data S2): C1 (Tm ∼82.95 °C; PRUS5_Moo and PRUS6_Moo; Fig. S3 and Data S2) and C15 (Tm ∼83.5 °C; PRUS3_Moo and PRUS4_Moo; Fig. S3 and Data S2). In subsequent analyses of the Symbiodinium community composition, each clade was expressed by 28S copy number per unit of coral 18S in order to cover clades A–F. The S/H ratio calculation displayed intra and inter-specific variation of the total Symbiodinium densities harbored within the host (Fig. 2A), either for a specific clade or from the total Symbiodinium density (all clade(s) included).

Figure 2 Quantitative composition of different Symbiodinium clades observed in association with ACYT: A. cytherea, APUL: A. pulchra, PCAC: P. cactus, PDAM: P. damicornis and PRUS: P. rus based on: (A) 28S copy number estimation (B) cell number estimation and (C) clade proportions within coral hosts.

The grey circles represent the presence of background clades under a 5% threshold (dashed line). Coral IDs are indicated under each histogram.

Table 1 Comparative census of Symbiodinium clades and types associated with common coral species from Moorea (A. cytherea, A. pulchra, P. damicornis, P. cactus, and P. rus) detected in a previous report by Putnam et al., 2012 vs. the present study.

Coral species	Previous report [1]	Present study	
	Clade(s)	Type	Clades	* Type	
A. cytherea	A, D	A1, D1	A, C*, D	C1*	
A. pulchra	A, D	A1, D1	A, C*, D	ND	
P. damicornis	A, C, D	DA, A1, C15	A, B*, C, D	B1*	
P. rus	C	C15	A*, C, D*	A13*, D1*, C15, C1*	
P. cactus	C	C1, C3, C45	C, D*	C1, D1*	
Notes.

* novel detected Symbiodinium clade/type for the listed coral species from this study.

ND no data

The occurrence of clades A, B, C, and D led to fifteen possible theoretical patterns among which nine have been observed previously, including assemblages of three clades together (Fig. 2): ACD (A. cytherea and A. pulchra), BCD (P. damicornis) or ABC (P. damicornis). However, Symbiodinium patterns that include clade B as either a unique clade (B) or as an additional clade (BA, BC, BD, BAD and ABCD) have never been recorded. Using the Symbiodinium densities (S/H ratio) within the coral host (Fig. 2C), relative proportions were determined, and allowed for their classification as either dominant (>5%) or background clade(s) (≤5%; Table 2). Symbiodinium clade B, only detected in P. damicornis, was always characterized as background regardless of the clade pattern (0.0002–0.0009% of the Symbiodinium communities; Table 2), and was systematically associated with at least clade C. All of the other three clades (A, C and D) were observed at least once as background clades, depending on the species and on the clade pattern. For example, clade A was occasionally found as background in P. rus with an AC-pattern (0.0001% within Li-05 and 0.002% within Va-03), and was frequently observed as background in A. cytherea (<2%; Table 2). Clade D was found as background in P. rus (0.026% within Va-05) or P. cactus (0.003% within Ti-05) with a CD-pattern. Clade C was observed as a background clade only once in P. damicornis with a CD-pattern (0.04% within Li-01). In some corals, different Symbiodinium clades occurred in more even proportions. For example, clades C (51.07%) and D (48.93%) within P. damicornis (Li-02; Fig. 2C) exhibited a BCD-pattern, and clades A (57.13%) and D (42.87%) showed AD-pattern within A. cytherea (Va-03; Fig. 2C).

Table 2 Proportion of background clades identified within the coral hosts A. cytherea, A. pulchra, P. cactus, P. damicornis and P. rus.

Species	Coral ID	Background clade proportion	
A. cytherea	Li-02	A = 0.0012%		
Li-03	A = 0.0005%		
Li-04	A = 1.5718%		
Va-01	A = 0.7750%		
Va-02	A = 0.1496%		
Va-04	C = 3.0797%	A = 0.2089%	
Va-05	A = 0.3314%		
Te-02	A = 0.0242%		
Te-03	C = 1.5921%		
Te-04	A = 0.1931%		
Te-05	A = 0.8460%		
Te-06	A = 0.7958%		
A. pulchra	Ti-04	C = 5.0116%		
Ti-05	A = 0.2073%		
Ti-06	D = 0.7418%	A = 0.3984%	
P. cactus	Ti-05	D = 0.0029%		
P. damicornis	Li-01	C = 0.0380%		
Li-02	B = 0.0002%		
Ti-01	B = 0.0009%	A = 0.0002%	
P. rus	Li-05	A = 0.0001%		
Va-03	A = 0.0020%		
Va-05	D = 0.0259%		

Selective coral-Symbiodinium partnerships

The discriminant analysis of principal components (DAPC; Fig. 3) on the five coral species showed compositional differences among associated communities of Symbiodinium according to clade identity and to their density in the host. The first axis (43.9% of total variance) of the DAPC opposed Symbiodnium communities characterized with higher clade D density (Pearson’s corelation: P < 0.001, t = 15.7) from communities composed of higher clade C (Pearson’s correlation: P < 0.001, t =  − 21.5) and/or clade B (Pearson’s correlation: P = 0.01, t =  − 2.5) densities. Clade D was strongly representative of P. damicornis Symbiodinium communities (100% of coral colonies sampled), nearly always appearing as a unique clade (24∕27 = 89%; Fig. 2). In contrast, P. rus (18∕21 = 85.7%; Fig. 2) and P. cactus (6∕7 = 85.7%; Fig. 2) colonies were nearly exclusively composed of mono-clade C communities. However, one P. cactus colony also associated with clade D (Fig. 2), underlying a larger range of variation in the density of the associated symbiotic communities (wide size of discriminant ellipse, Fig. 3). The second axis (24.9% of total variance) of the DAPC differentiated Symbiodinium communities was composed of clade A (Pearson’s correlation: P < 0.001, t = 11.4), and was comprised of both Acropora species. These two species mainly associated with multi-clade communitities (A. cytherea: 81% and A. pulchra: 67%) and were distinguished by a second preferential clade in addition to clade A (Figs. 2 and 3): D for A. cytherea (AD and ACD patterns 11∕16 = 68.8%) and C for A. pulchra (AC and ACD patterns 4∕6 = 66.7%).

Figure 3 Spatio-temporal multivariate analysis of clade A–D quantifications converted in 28S copy number.

Axis 1 and 2 of the discriminant analysis of principal component (DAPC) according to the five coral species: A. cytherea, A. pulchra, P. cactus, P. damicornis and P. rus.

Discussion

This study analyzed the Symbiodinium communities of five abundant coral species from Moorea (A. cytherea, A. pulchra, P. damicornis, P. cactus and P. rus), and found Symbiodinium clades A, C and D (from the six clades tested, A–F) in all of the species except for P. cactus, which was never observed in association with clade A. This is congruent with previous observations that have described these three Symbiodinium clades as the primary clades inhabiting scleractinian corals (Van Oppen et al., 2005). In contrast, while Symbiodinium clade B is commonly reported in Caribbean corals (Rowan et al., 1997; Diekmann et al., 2003; Pettay & Lajeunesse, 2007; Cunning, Silverstein & Baker, 2015), it is rarely reported in corals from the Central Pacific (e.g., LaJeunesse, 2001). This study is the first record of clade B found associating with corals from French Polynesia (see previous studies by Magalon, Flot & Baudry, 2007; Putnam et al., 2012). Clade B was detected exclusively as a background population in P. damicornis, and genotyped as belonging to sub-clade B1. Coincidentally, among the few detections of Symbiodinium clade B in Pacific corals to date (e.g., LaJeunesse, 2001; Silverstein, Correa & Baker, 2012; Parkinson, Coffroth & Lajeunesse, 2015; Lee et al., 2016), genotype B1 associated with P. damicornis in Hawaii (LaJeunesse, 2001). In addition, clade B has also been found in Moorea, but as a symbiont with the nudibranch Aeolidiella alba (Wecker et al., 2015).

The low abundances of clade B may also have come from an exogenous source (e.g., surface environmental cells), and therefore may represent a non-symbiotic interaction with the host. However, the strict conditioning of samples during DNA extraction (e.g., eliminating traces of mucus; described in Rouzé et al., 2016), and the absence of any detection of clade B in the other coral species from the same sampling site considerably reduce this hypothesis. Instead, the rarity and low abundance of B1 lineages in corals from Moorea may be consistent with a previous report in which a B1 type was found to opportunistically associate with Pocillopora colonies following a coral cold-bleaching event (LaJeunesse et al., 2010). However, a recent study by Lee et al. (2016) conducted in Republic of Korea found clade B (type B2) to commonly reside in the host tissues of Alveopora japonica. Alternatively, although Pacific corals rarely associate with clade B, the function of this symbiosis may represent an, as of yet, unknown ecological niche. However, given the rarity of this association, the significance of this partnership it likely to have minor physiological consequences on the host’s survival (e.g. sensitivity to thermal stress; Loram et al., 2007).

The qPCR assays revealed that each of the four clades A–D could be detected at least once at a background level (i.e., ≤5%), a finding that is consistent with previous studies (e.g., Mieog et al., 2007; Silverstein, Correa & Baker, 2012). In addition, this study increases the number of known background clades, and presents novel partnerships between corals and Symbiodinium (e.g., P. rus with clades A or D). However, some coral-Symbiodinium pairs were not recovered. For example, P. cactus was not found associating with clade A, and P. rus, P. cactus and the Acropora spp. did not associate with clade B. This could be due to a limited sampling effort among some of the corals (e.g., 6 A. pulchra sampled) rather than a selective exclusion by the host or symbiont to a particular partner by cellular recognition mechanisms (Silverstein, Correa & Baker, 2012; Davy, Allemand & Weis, 2012). While a majority of background clades were only occasionally detected within some coral species (e.g., clades A and D in P. rus or clade B in P. damicornis), the presence of clade A in low abundance in A. cytherea was nearly exclusive. Consequently, the ability of corals to harbour multi-clade Symbiodinium communities at background levels may be due to the environmental history of Moorea island, which has experienced a variety of massive bleaching events (Penin, Vidal-Dupiol & Adjeroud, 2012), and therefore represents a meaningful ecological function that could influence holobiont resistance (Berkelmans & Van Oppen, 2006; Mieog et al., 2007). Indeed, background clades support the potential for dynamic ecological strategies (e.g., switching vs. shuffling), as described in the ABH, that could lead to a rapid selective mechanism of tolerant coral-Symbiodinium partnerships in response to environmental change (Buddemeier & Fautin, 1993; Baker, 2003).

Despite the observed increase in variation among Symbiodinium clade associations for the five coral species studied, each species was restricted to a particular dominant or background Symbiodinium clade(s). For example, clade A was exclusively observed (>95%; Fig. 2) in the Symbiodinium assemblages associated with both Acropora species, either as a background clade or as a dominant clade. Similarly, other coral species exhibited a primary association with a particular clade: either exclusively, as with clade C in P. rus and clade D in P damicornis, or dominant, as with clade C in P. cactus. Similar to the study by Putnam et al. (2012), which investigated a wide range of fringing reefs during different seasons (i.e., dry season in this study vs. April: wet season), our findings also support consistent associations of coral species to particular Symbiodinium clades (LaJeunesse et al., 2008; Stat et al., 2009). Given our standardized sampling method, the few exceptions of the multi-clade associations found for P. damicornis or P. cactus could be preferentially attributed to a transient acquisition of Symbiodinium clades (Muscatine, 1973; Yamashita et al., 2011; Lee et al., 2016), rather than spatial partitioning of Symbiodinium within host colonies (e.g., Rowan & Knowlton, 1995). Overall, our findings are consistent with corals as ‘specialists’ (low flexibility: specific to particular symbiont(s)) or ‘generalists’ (high flexibility: associated with various symbionts). To further explore symbiont diversity in corals, similar fine-scale molecular approaches (e.g., qPCR, next generation sequencing; see Barbrook, Voolstra & Howe, 2014) should be performed on a wide range of coral species throughout a large geographic range.

The association of coral species with specific clade(s) observed among Moorea’s reefs is consistent with previous reports of stable partnerships between coral hosts and subsets of Symbiodinium (e.g., Thornhill et al., 2006; Thornhill et al., 2009; Suwa, Hirose & Hidaka, 2008; Rouzé et al., 2016). Such symbiotic specificity can be derived from the ‘winnowing’ of multiple symbiont types initially present in the host (Nyholm & Mcfall-Ngai, 2004). This process occurs in many mutualistic relationships (e.g., legume-zhizobial bacteria: Hirsch, Lum & Downie, 2001; squid-luminous bacteria: Nyholm & Mcfall-Ngai, 2004; cnidarian-Symbiodinium: Wolfowicz et al., 2016), and consists of a complex series of molecular recognition interactions between the host and the symbionts. It is likely that the specific partnerships observed between corals and Symbiodinium are derived from various biological traits (Yost et al., 2013), as well as different physiological and ecological attributes among Symbiodinium clades (Kinzie et al., 2001; Berkelmans & Van Oppen, 2006; Hennige et al., 2009; Baker et al., 2013) that ultimately lead to the ecological success of distinct holobionts. For example, the high resistance of Porites to a variety of stressors could be explained, in part, by its stable association with Symbiodinium type C15 (Putnam et al., 2012). This symbiont has been characterized as thermally tolerant (LaJeunesse et al., 2003; Fitt et al., 2009) and more resilient to extreme environmental conditions compared to other clade C types (LaJeunesse et al., 2003), which may have favored its ecological radiation throughout the Indo-Pacific (LaJeunesse, 2005; Pochon et al., 2007). In this study, and similar to a previous report by Putnam et al. (2012), P. cactus always associated with Symbiodinium in clade C. However, it likely belongs to type C1 (Fig. S3 and Data S2; Putnam et al., 2012) which is described as thermo-sensitive (Deschaseaux et al., 2014), and that could explain the lower resistance to environmental conditions of the species. Similarly, the ecological sensitivity of branching corals from the genera Acropora and Pocillopora could be explained, in part, by their specialization with Symbiodinium clade A (type A1; Putnam et al., 2012 and type A13), and clade D, type D1/D1a (Putnam et al., 2012) respectively. Indeed, while both clades A and D are often linked with eco-physiological benefits for the holobiont, including photo-protective and thermo-tolerance abilities, respectively, they have mainly been described in stressful vs. non-stressful conditions. This duality likely corresponds to trade-offs between coral host resistance and low energetic budget contributions (reviewed in Lesser, Stat & Gates, 2013). In some cases, Symbiodinium belonging to these clades have been reported as nominal contributors to host metabolism (e.g., growth and reproduction (Little, Van Oppen & Willis, 2004; Jones & Berkelmans, 2010) and/or nutrition (Stat, Morris & Gates, 2008; Cantin et al., 2009; Baker et al., 2013)).

The specialization of coral hosts to particular Symbiodinium clades likely represents a driver resulting in stable mutualisms, initiated from selective pressure, that enhances the benefits of specific symbiosis through co-evolution (Douglas, 2008; Thornhill et al., 2014). However, this specialization is contrasted with the maintenance of the horizontal transmission of symbionts in the majority of coral species as well as the detection of additional clades, at trace levels, within the five coral species examined. Lee et al. (2016) suggest that low abundance ‘background’ Symbiodinium populations are not necessarily mutualistic but can reflect a transient relative abundance in the surrounding environment, such as non-directional ingestion by polyps leading to ephemeral symbiont shifts (LaJeunesse et al., 2009; Stat et al., 2009; Coffroth et al., 2010). Nevertheless, every Symbiodinium species may not be transiently ingested. For example, Symbiodinium clade F was never found in the host tissues of the five coral species examined, despite clade F being detected in the surrounding environment (Fig. S4), and, although in a temperate environment, described as a dominant symbiont within Alveopora japonica (Lee et al., 2016). This suggests a combination of physiologically controlled processes among the coral host and its background Symbiodinium communities. Therefore, two opposite selection pressures may be co-occuring in the context of Moorea’s reef environment (which has been exposed to consecutive massive bleaching events in the past): (i) the optimization of a symbiosis with a specific clade(s) and/or (ii) the maintenance of the ability to integrate several different (but not all) clades in low abundance that could yield an overall benefit to the coral holobiont.

Altogether, these findings emphasize the need to better understand whether those Symbiodinium present in low abundance play an ecological role for the holobiont over time, and to further explore the processes that may govern the maintenance of Symbiodinium in low abundance in addition to the dominant symbioses that occur with particular clades.

Supplemental Information

Supplemental Information 1 Supplementary methods

Click here for additional data file.

Data S1 Raw datas: qPCR results

Click here for additional data file.

Figure S1 Efficiency of specific-clade primer sets

Standard curves for different primers, corresponding to Ct values versus logarithmic 10 fold dilution of purified 28S PCR products. Each plot corresponds to individual Ct values obtained from three technical replicates. Primer efficiencies were deduced by linear regression: (A) mix of purified 28S PCR products from Symbiodinium A–F each concentrated at equal concentration, 1⋅1011×(−0.681x) (clade A, R2 = 0.999), 1⋅1011 × e(−0.676x) (clade B, R2 = 0.999), 1⋅1012 × e(−0.687x) (clade C, R2 = 0.983), 1⋅1012 × e(−0.694x) (clade D, R2 = 0.998), 6⋅1010 × e(−0.663x) (clade E, R2 = 0.998) and 8⋅1010 × e(−0.694x) (clade F, R2 = 0.997); and (B) mix of coral DNA each concentrated at equal concentration, universal coral primer set, 1.25⋅104 × e(−0.694x) (mix of 10 coral species: R2 = 0.999), 9.02⋅105 × e(−0.705x) (mix of A. cytherea: R2 = 1.000), 6.58⋅105 × e(−0.692x) (mix of P. rus: R2 = 0.999) and 1.0⋅106 × e(−0.629x) (mix of P. damicronis: R2 = 0.996).

Click here for additional data file.

Figure S2 Efficiency of specific-clade primer sets on Symbiodinium cells isolated from corals

Standard curves for clade-specific primer sets for clades A, C and D corresponding to cell densities versus logarithmic 10 fold-dilution of isolated coral-symbiotic Symbiodinium. Each plot corresponds to individual Ct values obtained from three technical replicates. Primer efficiencies were deduced by linear regression: 1⋅1010 × e(−0.628x) (clade A, R2 = 0.85), 1⋅108 × e(−0.45x) (clade C, R2 = 0.91) and 4⋅106 × e(−0.363x) (clade D, R2 = 0.92).

Click here for additional data file.

Figure S3 Phylogenetic tree of Symbiodinium using sequences of 28S rDNA

Phylogenetic tree of Symbiodinium clades A-F derived from bayesian analyses using sequences of 28S rDNA from Moorea (Moo; in black bold; sequences available in supplementary data: dataset 3), BURR collection strains described in Table S2 (in grey bold), and published genetic sequence from GenBank (regular font). Bayesian posterior probability (in percentage; first value) and Maximum Likelihood bootstrap support values (second value) are presented following the substitution model of Kimura 2-parameter with a proportion of invariable sites.

Click here for additional data file.

Figure S4 Symbiodinium clades C and F in seawater samples of Moorea

Presence of Symbiodinium clade C, E and F in seawater samples of Moorea. PCR amplifications of 28S rDNA using the primer sets clade-specific C, E and F (Yamashita et al., 2011) on DNA extracts from saturated 0.2 µm filters (filtration volume: 6-9 L) of 4 seawater samples (M1, M2, M3 and M4): T−: negative control (no DNA); TD: positive control (clade D DNA). A 100bp amplicon characterizes a positive amplification. Uppercase letters indicate the corresponding clade.

Click here for additional data file.

Table S1 Characteristics of primer sets

Click here for additional data file.

Table S2 qPCR assays on cultures Symbiodinium strains

Values of Ct obtained by qPCR assays of the corresponding clade-specific primer sets on different DNAs issued from cultured Symbiodinium strains (clades A to F; BURR collection). Each DNA was tested at an equal amount of 10 ng per reaction. (-) means no amplification or unspecific amplifications based on dissociation curves analysis, and (*) symbiotic strains used in mix at same concentration and tested with each primer set.

Click here for additional data file.

Data S2 Symbiodinium 28S DNA sequences mentioned in the paper

Symbiodinium 28S DNA sequences mentioned in the paper and used in the phylogenetic analysis (phylogenetic tree: Figure S3).

Click here for additional data file.

We thank B Espiau and M Vairaa from CRIOBE and J Fievet (in charge of qPCR platform utilization) from IFREMER from Taravao for their advices and assistance in laboratory. We are also grateful to MA Coffroth and her laboratory (BURR; http://www.nsm.buffalo.edu/Bio/burr/) for providing us cultured Symbiodinium strains. We also thank C Bonneville who performed analyses of Symbiodinium diversity in the water column during her master internship.

Additional Information and Declarations

Competing Interests

Author Contributions

DNA Deposition

Data Availability

The authors declare there are no competing interests.

Héloïse Rouzé conceived and designed the experiments, performed the experiments, analyzed the data, wrote the paper, prepared figures and/or tables, reviewed drafts of the paper.

Gaël J. Lecellier conceived and designed the experiments, analyzed the data, contributed reagents/materials/analysis tools, reviewed drafts of the paper.

Denis Saulnier, Serge Planes and Yannick Gueguen contributed reagents/materials/analysis tools, reviewed drafts of the paper.

Herman H. Wirshing reviewed drafts of the paper.

Véronique Berteaux-Lecellier conceived and designed the experiments, contributed reagents/materials/analysis tools, reviewed drafts of the paper.

The following information was supplied regarding the deposition of DNA sequences:

Sequences have been provided in GenBank (KY110998 –KY111024) and as Data S2.

The following information was supplied regarding data availability:

The raw data has been supplied as Data S1.

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
