# Peer review of "An updated assessment of Symbiodinium spp. that associate with common scleractinian corals from Moorea (French Polynesia) reveals high diversity among background symbionts and a novel finding of clade B"

_PeerJ, doi:10.7717/peerj.2856_

## Round 0.1 · original submission · Major Revisions

· Academic Editor

Major Revisions

All three reviewers have carefully reviewed your manuscript and have consistently highlighted some significant issues with the overall quality, the methodology, the presentation and reporting of results within the current knowledge of the field, and the discussion/interpretation of the data. Each reviewer has provided you with some excellent suggestions (see attached annotations) for improving both the methodology and the quality/direction of the next manuscript.

All reviewers also insisted that, considering the methodological approach/issues and the rather limited sample size used, the manuscript is currently pushing for a much larger story than the presented data can actually support. They acknowledged that the primary novelty of the study is the description of Symbiodinium communities in a relatively unexplored region of the world and the finding of Symbiodinium clade B for the first time in Moorea. Because neither the concept of ‘Symbiodinium clades/types faithfulness’ nor the qPCR approach used are new (considerable literature already exist), reviewers advocate for a major shift in focus from ‘functional faithfulness’ to ‘reporting of Symbiodinium diversity in Moorea’, and to more appropriately discuss the data within the current context of the field.

I agree with their assessments and advise you to address each point carefully; In particular, I advise you to:

1) Modify the current title of the manuscript, excluding the concept of faithfulness. Perhaps, incorporating the new finding of clade B?
2) Provide a clear hypothesis in the introduction.
3) Fully clarify the qPCR methodological issues and provide the actual raw data, as requested by reviewers 1 and 2.
4) Rewrite/Refocus the introduction and discussion sections of the manuscript as described above.
5) Significantly improve the overall quality of the paper, including acceptable English grammar and sentence structure, appropriate representation of published work, and properly formatted references. Currently, the manuscript does not meet PeerJ standards. This could be facilitated through the invitation of one of your English-speaking colleagues as an additional co-author.

·

Basic reporting

Many of the ideas presented in the introduction and discussion are vague and ambiguous (e.g., line 48: “coral-algal association can be broken”; line 304: “low beneficial physiological properties for host survival”; line 327: “non-respected privileged partnerships” and “punctual dynamic lost [sic] of Symbiodinium clade”; line 364: “the strength and efficiency of a specific symbiosis” and the “development of new cellular processes”, and many more examples). Overall, the writing should be refined and sharpened with greater precision and detail, and could also benefit from greater consistency with language used in published literature (e.g., the concept of a “faithful clade” is well described in terms of “specificity”, etc.).
There are some critical gaps in the methodology and reporting of results as well. For example, the sequences derived from sequencing of amplified products corresponding to “new” clade associations are not povided, nor is it described how these sequences were assigned a taxonomic identification. The legend for Table S1 also requires a detailed description explaining the information presented in this table. For example, what is the definition of “Sensitivity Ct” and “Copy number threshold”, and how were these values derived?

Experimental design

The methodology for quantifying Symbiodinium is very unclear. In particular, I do not understand the process of “conversion to 28S copy number” for “absolute quantification of each Symbiodinium”. A much clearer description of these methods and rationale is needed. One major concern is primer specificity. Table S2 shows that several of the primer sets amplify non-target clade DNA to some extent. It is stated that melting curve analyses ensured the specificity of amplifications, but melting curve temperatures do not appear very different among clades and this could be a potential source of error.
Another major concern is that there is no attempt to account for differences in 28S gene copy number among different Symbiodinium clades (e.g. the number of copies of the target locus per Symbiodinium cell), which may make the values of proportions of each clade within a sample meaningless. These copy number values are also likely to vary within clades, therefore even an attempt to control for this at the clade level is problematic, and confounds comparisons of relative clade abundances across coral species. Analyzing these data as presence-absence, or as “order-of-magnitude” differences (e.g., Correa et al. 2009) may be more appropriate.
It is also not clear why a coral 18S assay was necessary, since the data are only presented as relative abundances (i.e., proportions) of different Symbiodinium clades within samples. These values can be derived without normalizing to host DNA; see the attached table using hypothetical data for a single coral sample. This table shows that the proportions of each clade within a sample can be determined using the ratios of each clade to the most abundant clade, with no need to normalize each clade to the amount of host DNA. If this paper only analyzes the proportions of clades within samples, then the methods could be greatly simplified by removing the host quantification altogether.
Finally, it is not entirely clear how the inter-plate calibrator and the “estimated threshold ranges” or “limit of quantification” were used to correct Ct values and score positive amplifications. Providing the actual raw data (Ct values) along with a spreadsheet or R script showing each step along the way to arrive at the final proportion data (i.e., the “interplate correction”, “sensitivity thresholding”, “28S conversion” steps, etc.) would make this work much clearer and more reproducible.
In addition to providing the raw data, the authors should make sure to describe the data in detail. For example, with the currently provided file, it is not clear what the columns named “Col” , “Nbcol”, and “Comb” represent.

Validity of the findings

Validity of these findings will be much easier to assess once the methods of symbiont quantification and data analysis have been presented more clearly. However, notwithstanding these improvements, the attempt to analyze and discuss clade associations in relation to differences in coral morphology, symbiont transmission mode, and environmental sensitivity is weak, and probably not possible with only 5 coral species. It may be more straightforward to frame the overall approach as a gross characterization of symbiont clade associations detected by qPCR within dominant corals in Moorea, rather than trying to parse differences among different coral functional groups – the dataset is not large enough to make these distinctions.

Additional comments

Describing the diversity of associations between coral species and various Symbiodinium types is an important research direction, and this work represents a potentially valuable contribution to this effort. However, gaps in the description of the methodology and the lack of transparency in analysis (and lack of raw data) make the validity of these findings difficult to assess. The writing also suffers from ambiguity and imprecise language, and major rewriting and potentially reframing of this work is necessary.

Reviewer 2 ·

Basic reporting

In their article entitled ‘Diversity of cryptic Symbiodinium clades in common coral genera from Moorea, French Polynesia: faithfulness does not exclude flexibility’, Rouzé et al. apply a Symbiodinium-specific qPCR assay to five common coral species, and report the presence of multiple symbiont clades at biologically relevant abundance in several of the samples. The authors conclude that two opposing processes are responsible for the mixed symbiont assemblages: host-symbiont fidelity and the need to maintain flexibility in the face of changing environmental conditions. The novelty and scope of this study are rather limited, since a moderate number of corals were analysed at a single time point, using a qPCR assay from a previous study (Yamashita et al. 2010, Marine Biology 158:87-100). Several reports of multiple clades coexisting within individual colonies appear in the literature; therefore the primary novel finding of this study appears to be the occurrence of clade B Symbiodinium in association with scleractinian corals at this location.

This article is currently in a poor state of completeness, and substantial editing is required to bring it up to PeerJ standards. This is immediately obvious from the first sentence of the abstract, which contains at least four grammatical errors (see notes in the attached pdf file). There also doesn’t appear to be any clear hypothesis/research question stated. Rather the last part of the introduction consists of sentences that belong in the methods and results sections. The reference section contains multiple errors and incorrect formatting, with none of the genus/species names italicized, journal names are not capitalized (e.g. Molecular biology and evolution page 469), capitalization in article titles is inconsistent, some author names are misspelled (e.g. ‘Hoeghguldberg’ line 556), and some author’s initials are in lower case (I understand that this is an annoying feature of some reference manager software, but these should have been checked and corrected prior to submission). GenBank reference numbers throughout the manuscript are missing (e.g. line 111: Genbank reference xx-xx), showing a further lack of completeness. Before accepting this article for publication I would expect the authors to perform a thorough check and correction of grammar, spelling and formatting throughout the article.

Experimental design

The incomplete state of the article made it rather difficult for me to comment on the robustness of the scientific process. The qPCR validation appeared to be correct for the most part; however I have a concern about Lines 151-158, which outline the multi-clade DNA mix used to test primer efficiency and specificity. Were the Symbiodinium cell suspensions added at equal cell concentrations (i.e. cells per µl) or at equal DNA concentrations (i.e. ng per µl)? This is important because the per-cell rDNA copy number can vary widely between different Symbiodinium clades (Meiog et al. 2007, Coral Reefs 26: 449-457 ) and subclades (Wilkinson et al. 2015, BMC Evolutionary Biology 15:46). If this DNA mix was comprised of equimolar DNA concentrations it is impossible to accurately enumerate symbiont abundance ratios. This limitation would need to be explicitly stated in the discussion section.

Validity of the findings

See comments below and notes in the attached pdf document:

-The use of the word ‘stability’ throughout the text implies that the association does not change through time; however this factor was not explicitly tested. There is therefore no guarantee that the association is stable, and not a transient infection.

-On Lines 104-105, are you claiming to have formulated a new concept of a ‘faithful clade’? It is well known that corals co-evolve with certain symbiont lineages, often referred to as homologous zooxanthellae or symbionts (sensu Davy et al. 1997 Biological Bulletin 192:208-216), and usually involving a particular sub-cladal type (e.g C15 with Porites, etc). Is your finding that host identity predicts symbiont clade better than sub-clade? If so this isn't clear.

-A phylogenetic network indicating the relationship of your new sequence to the others in this study and others would be a valuable addition.

-It may be worthwhile noting that Symbiodinium minutum (the B1 type that is homologous with the anemone Aiptasia) can grow opportunistically following coral bleaching (LaJeunesse et al. 2009 Proc. B 276:4139-4148), presenting a possible explanation for its presence at low abundance in some of these Pocillopora colonies.

-I believe the authors should make their raw Ct values available, either as supporting information or through an online repository such as Dryad.

Additional comments

No additional comments

Annotated reviews are not available for download in order to protect the identity of reviewers who chose to remain anonymous.

·

Basic reporting

While I respect the work and efforts of these authors, the manuscript requires major editing throughout to clarify the intended message. Unfortunately, this goes beyond a basic grammatical check to include larger sections that lack clarity and the misuse of English words throughout which create ambiguity in meaning. With that said, the contribution of a more complete description of Symbiodinium communities within Moorea is significant and noteworthy. The incidences of high within colony symbiont diversity are unique and interesting.

Experimental design

The manuscript includes data for an impressive number of host species throughout Moorea and found many incidences of previously unidentified symbiont diversity. The sampling effort may have benefited from a higher sample size of some of host species (i.e., where 6-7 individuals colonies were sampled) as the manuscript emphasizes the need to high sampling to detect seemingly rare occurrences of certain host-symbiont pairings. The development of new qPCR targets that are robust at the clade level is an important contribution to the field. There were some sections of the methods that were difficult to understand due to the aformentioned language issues and thus my comments on this section are limited and I will default to editors with specific experience in developing qPCR assays. I do have a general concern about the lack of detection of clades at very low relative abundances (less than 1%) as has been so commonly reported with other qPCR methods (e.g. many works by Silverstein, Cunning, Mieog etc.). It is possible that this does not occur within Moorea but that argument would be strengthened by trials on mixed culture dilutions for example. Unfortunately, as I have experienced myself, new methods tend to have a high burden of proofing and ground truthing.

Validity of the findings

There is quite a bit of interesting information within the manuscript, however, the framework within which it is presented has faults. I have included specific notes within the pdf, particularly within the discussion. I also feel that there is some unclear representation, perhaps unintentional, of published works, and a lack of inclusion of existing works. I have also included specific notes on within the pdf. I have done my best to address these areas in a way that can be constructive for the authors and I apologize in advance if there is any unintended coarseness.

Additional comments

As I have mentioned the data are interesting, however I find that the manuscript, in particular the discussion, attempts to push a larger/different story than what those data can support. I find it quite interesting and somewhat novel that the clades can be so evenly distributed within some colonies. A discussion with of a more inclusive description of the factors that may drive these "exceptions" to mono-specific (or almost mono-specific) symbiont communities found elsewhere would be an excellent contribution to the field. There is quite a bit of literature that could be presented in this light. As currently presented, I cannot recommend this manuscript to be accepted for publication as is. As I do believe that the content could be reworked and presented in a different context, I have spent considerable time to include specific notes within the PDF that I hope will encourage and aid the authors in getting this information out into the literature.

---

## Round 0.2 · Minor Revisions

· Academic Editor

Minor Revisions

Dear Heloise and collaborators,

Thank you very much for your recent resubmission. As evidenced by the comments from both reviewers, your manuscript has greatly improved and I thank you for all the good effort you've put into it. Both reviewers (see below) have added a number of excellent suggestions, which I recommend should be carefully integrated into your final manuscript. Also note the edited PDF from reviewer 2 containing edits and comments.

Please provide me with a point-by-point response and modified manuscript, and I will then be happy to consider it positively for publication.

All the best,
Xavier

Reviewer 2 ·

Basic reporting

The revised manuscript represents a significant improvement over the originally submitted version. The majority of reviewer comments have been addressed. Minor further revisions follow in the general comments section.

Experimental design

No Comments.

Validity of the findings

No Comments.

Additional comments

The presence of clade B Symbiodinium is presented as a key finding in this study; however, it was detected at very low abundance in only two Pocillopora colonies. The copy numbers from the qPCR spreadsheet were only 44 and 161, with very high CT values in the range of those that were assigned as ‘negative’. The qPCR products were sequenced and found to contain clade B. However, it is possible that this was caused by PCR contamination. This possibility should be addressed in the text.

Some discussion on the biological relevance of Symbiodinium that occur in such low abundance is warranted. For example, certain types that were detected at trace levels could simply represent surface environmental cells that have no intracellular interaction with the host, and are unlikely to be involved in switching/shuffling.

Line 238: R package ade4 needs to be cited, as does R itself. Were the t-tests and correlations done in R too?

Lines 350 – 352: There are many reports of clade B Symbiodinium in corals, including the one cited in the next section (LaJeunesse et al. 2009). Many soft corals host clade B, and a recent paper by Parkinson et al. (2015; J. Phycol 51:850-858) described several new clade B species, including which host species they were isolated from.

Lines 355 – 357: I’m not sure that LaJeunesse et al. (2009) considered Symbiodinium B1 in S. siderea as ‘opportunistic’. Perhaps a better example is from LaJeunesse et al. (2010) Proc Roy Soc B. 277: 2925-2934, where a B1 strain (probably S. minutum) grew opportunistically on cold-bleached Pocillopora colonies.

Line 419: Change ‘it is likely belonging’ to ‘it likely belongs’

There are still reference formatting problems. For example, on line 575 the Muscatine reference is missing the journal volume. Also, the capitalization of journal titles (such as PLOS) and author names (e.g. van Oppen) is inconsistent throughout the bibliography. These should be checked carefully prior to publication.

·

Basic reporting

The context of the manuscript is greatly improved and better represents its contribution to the field as a paper describing new diversity within coral-Symbiodinium associations in Moorea, French Polynesia, which to me is quite interesting. The English is also greatly improved from the previous version. There are still some errors, some of which I addressed in the PDF but also others so that it should be carefully reviewed before final publication. Most of these are minor but some areas could use further clarification (noted within PDF). I have suggested further citations in a few places within the PDF.

Experimental design

The design is a robust sampling from around the island. While I am not the foremost expert in the design of new qPCR assays, I still have some concerns with how the quantification is presented. Presenting the 28S vs 18S ratio works for the relative abundances of symbiont communities as described in the paper (Fig. 2c). I am less comfortable with the quantification in Fig. 2a, 2b as without a 28S/18S per cell of the symbiont/host, respectively: we can't quantify the actual symbiont densities per host cell to compare to other qPCR work (e.g., Cunning and Baker 2013, and others) though they are presented similarly. I recommend a sentence or two that is explicit about the difference between the two evaluations of S/H ratio. However, the authors do a good job of drawing their major conclusions from the relative proportions which I do believe are accurate.

Validity of the findings

The finding seem to be valid and as mentioned in the previous review are quite interesting and novel. The major revisions have address the majority of my previous concerns. There remain a few weaknesses within the discussion section that could benefit from some re-writing and clarification. Most importantly, since the focus on the paper is on coral-symbiont associations within Moorea, the findings need to be put in better context as to the environmental conditions of Moorea. The conclusions drawn from comparing results from this work to trends seen in very different environments, e.g., Jeju (Lee et al. 2016), without explicitly discussing what is happening in Moorea seems inappropriate and a missed opportunity to focus on what actually might drive patterns in this newly presented Moorea data set. Perhaps revisit Penin, Vidal-Dupiol & Adjeroud, 2012 findings as to bleaching patterns and this data set. I appreciate the authors' efforts to put things into "big picture" concepts, (the ABH, co-evolution of host and symbiont, etc.) but I feel that presented data would be more appropriate, and have more value, within more of a Moorea focused context.

Additional comments

The major re-writing of the manuscript undertaken by the authors has greatly improved and clarified their major findings. Overall, with revisions I think the manuscript should be recommended for publication.

---

## Round 0.3 · Minor Revisions

· Academic Editor

Minor Revisions

Dear Heloise,

Thank you very much for your manuscript. I am very pleased with the way you have addressed the second round of reviews. I have made a number of comments/edits in the annotated pdf file (attached), for your considerations. Also, were you able to submit your Symbiodinium sequences to GenBank? If not, please make sure your MS refers to those sequences in the Supplementary File.

Once these have been incorporated, I will accept your manuscript for publication in PeerJ. Thank you for your hard work!

Xavier

---

## Round 0.4 · accepted · Accept

· Academic Editor

Accept

Dear Heloïse,

I am pleased with the last modifications you made. This publication will make a nice contribution to the field. Thank you for your hard work and patience in this relatively ling review process.

With kind regards,
Xavier